# Adversarial Mask Explainer for Graph Neural Networks

Affiliation Address
xxx@xxx

## ABSTRACT

The Graph Neural Networks (GNNs) model is a powerful tool for integrating node information with graph topology to learn representations and make predictions. However, the complex graph structure of GNNs has led to a lack of clear explainability in the decision-making process. Recently, there has been a growing interest in seeking instance-level explanations of the GNNs model, which aims to uncover the decision-making process of the GNNs model and provide insights into how it arrives at its final output. Previous works have focused on finding a set of weights (masks) for edges/nodes/node features to determine their importance. These works have adopted a regularization term and a hyperparameter $K$ to control the explanation size during the training process and keep only the top-$K$ weights as the explanation set. However, the true size of the explanation is typically unknown to users, making it difficult to provide reasonable values for the regularization term and $K$. In this work, we propose a novel framework AMExplainer which leverages the concept of adversarial networks to achieve a dual optimization objective in the target function. This approach ensures both accurate prediction of the mask and sparsity of the explanation set. In addition, we devise a novel scaling function to automatically sense and amplify the weights of the informative part of the graph, which filters out insignificant edges/nodes/node features for expediting the convergence of the solution during training. Our extensive experiments show that AMExplainer yields a more compelling explanation by generating a sparse set of masks while simultaneously maintaining fidelity.

## CCS CONCEPTS

• **Computing methodologies** → **Neural networks**; • **Mathematics of computing** → **Graph algorithms**; • **Security and privacy** → **Information-theoretic techniques**.

## KEYWORDS

Graph Analysis, Graph Neural Networks, Explainability

**ACM Reference Format:**
Anonymous Author(s). 2018. Adversarial Mask Explainer for Graph Neural Networks. In *Proceedings of Make sure to enter the correct conference title from your rights confirmation emai (WWW)*. ACM, New York, NY, USA, 9 pages. https://doi.org/XXXXXXX.XXXXXXX

## 1 INTRODUCTION

Graph Neural Networks (GNNs) have shown superior performance in handling irregular data compared to traditional neural networks. This is particularly evident in domains such as social networks [36] and chemical molecules [22]. By leveraging both node features and structural information, GNNs can effectively learn graph representations, which in turn can be utilized in various tasks such as node classification [14, 34, 9, 38], link prediction [44], and graph classification [5]. However, similar to other neural networks [15, 10, 17], the lack of explainability hinders the use of GNNs, especially in domains that value transparency, fairness, and safety. To address this limitation, it is necessary to provide explanations for the predictions made by GNNs. This requires identifying the most informative parts of the original graph. However, due to the complex aggregation of node features and graph topology, classical explanation methods [16, 19, 8, 4, 31] are not suitable for GNNs, making it challenging to understand the reasoning behind GNN predictions. Figure 1 (a) illustrates an example of Graph Neural Networks (GNNs) explainability applied to node classification. In this example, the purpose of the GNNs model is to determine the class to which node $v$ belongs, without revealing to the user the basis on which the GNNs model made this prediction. (In this case, the binary classification of a node is determined by whether the node is part of a subgraph resembling a house shape, as indicated by the dot dashed lines in Figure 1 (a).) The goal of the GNNs explainer, on the other hand, is to elucidate why the GNNs model made this prediction. In other words, if the GNNs model predicts that node $v$ belongs to a subgraph resembling a house shape, the purpose of the GNNs explainer is to identify and label this house-shaped subgraph in the graph.

GNNExplainer, the first explainability method for GNNs [39], uses soft masks on edges or node features to determine their importance. These masks, treated as trainable parameters, are combined with the original graph through element-wise multiplications. The training process involves maximizing the mutual information between the prediction made by the original graph and the explanation. To prevent the trivial solution where all masks have a value of 1 (because the entire graph is a valid but trivial explanation), a regularization factor is used to suppress the sum of all mask values. Once the mask values converge, the top $K$ edges/node features with the highest mask values are selected as the explanation set. Other works [20, 35, 26] have adopted similar ideas to generate a sparse explanation set. **However, suppressing mask values using a regularization factor during the training process may not be reasonable since the true size of the explanation set is unknown. Additionally, over-reliance on a manually set hyperparameter $K$ based on prior knowledge also diminishes the practicality of this model.**

In this paper, our focus is on finding post-hoc instance-level explanations for GNN predictions by identifying a small set of the

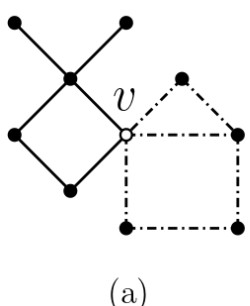 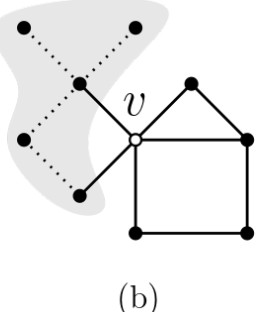 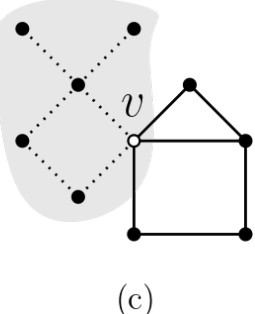

(a)  (b)  (c)

**Figure 1: (a) Example graph for node classification GNNs: to determine whether node $v$ is in a *house* subgraph. A trained GNNs model aims to tell users the answer yes or no, while a GNNs explainer targets to find such a *house* subgraph to explain why the GNNs model makes this decision. (b) AMExplainer identifies an explanation edge set (solid lines) whose complement set consists of uninformative edges (dotted lines in the gray area). (c) By employing a scaling function, AMExplainer further discovers and removes additional uninformative edges to obtain a sparser explanation set.**

most informative edges. We aim to address the aforementioned existing issues in this area of research, and our work is based on two key observations: (1) We note that the masked graph can be used as an input to approximate the original graph's prediction. Thus, the value of $mask \in [0, 1]$ is a measure of the extent to which the selected edges/nodes/node features contribute to the prediction. (2) We observe that the mask complement, with a value of $1 - mask$, is a measure of the extent to which the selected edges/nodes/node features contribute to the ambiguity of the prediction, or false prediction. Therefore, we expect the input with mask complement have no predictive ability on any class. Building upon these observations, we propose a novel framework called AMExplainer (Adversarial Mask Explainer) that trains an adversarially inspired network between the mask containing correct information and the mask complement containing incorrect/redundant information to generate an explanation with high sparsity instead of directly suppressing the size of the explanation set. Furthermore, we introduce a novel scaling function that effectively detects and enhances the mask weights of informative elements within the graph. This function serves to filter out irrelevant edges, nodes, and node features, thereby accelerating the convergence of the training process towards a solution. For instance, in Figure 1 (b), AMExplainer approximates an uninformative uniform distribution for the classification of node $v$ by making the complement set of the explanation edge set (indicated by solid lines, while the complement set is marked by dotted lines) approach this uniform distribution. It identifies a subset of edges (highlighted by the gray area and dotted lines) that do not contribute any information to the classification of $v$. To find more uninformative edges and make the explanation edge set sparser, AMExplainer employs a scaling function during training. This function speeds up the convergence of mask weights for uninformative edges, making the final explanation edge set closer to the ground truth, as shown in Figure 1 (c).

The contributions of this work are summarized as below:

- We have studied the problem of finding instance-level explanations for GNNs. We have observed that the mask complement is a measure of the extent to which the selected edges/nodes/node features contribute to the ambiguity of the prediction or false prediction.
- We have proposed a novel framework AMExplainer which leverages the concept of adversarial networks to achieve a dual optimization objective in the target function. This method ensures accurate predictions when using the identified explanation set as input for GNNs, while maintaining the sparsity, which enables a clearer understanding of how the explanation set contributes to GNNs' predictions.
- We have devised a novel scaling function to automatically sense and amplify the weights of the informative part of the graph, which filters out insignificant edges/nodes/node features for expediting the convergence of the solution during training.
- We have conducted extensive experiments on benchmark datasets to demonstrate that, while ensuring the sparsity of the explanation set remains comparable to the state-of-the-art, the explanation sets generated by AMExplainer significantly enhance prediction effectiveness by reducing the fidelity by $79.93\% - 94.22\%$ in both node classification and graph classification tasks.

The remainder of this paper is structured as follows. Section 2 provides a literature review of the relevant works. Section 3 introduces the preliminaries and our methodology. Section 4 shows the results of the experimental study. Finally, Section 5 gives our conclusion.

## 2 RELATED WORKS

**Graph Neural Networks**: GNNs have gained widespread popularity in recent years due to their capacity to learn representations from graph data, which can be applied to diverse downstream tasks, including node classification [14, 34, 9, 38], link prediction [44], and graph classification [5]. Some GNNs [14, 33, 3] are based on graph spectral theory and aim to apply convolutional operations to graphs. The majority of GNNs [34, 6, 1, 37] employ a similar

message-passing framework, where the node representation is updated by iteratively aggregating neighbor information. The differences among them lie in how to compute messages between nodes, how nodes update themselves with the received messages from neighbors, and so on. The effectiveness of GNNs in diverse downstream tasks can be attributed to their capacity to use both node information and graph topology simultaneously. However, this also presents a challenge in elucidating their internal workings, which is the main target of this work.

**Perturbation-Based GNNs Explanation Methods**: Perturbation-Based GNNs Explanation Methods aim to maximize the mutual information between explanation and prediction by generating masks for edges, nodes, or node features. Our work falls into this category. GNNExplainer is the first method of this kind, generating input-dependent explanation for GNNs. Graphmask [26] and PGExplainer [20] are subsequently proposed, with PGExplainer learning edge masks by training a neural network to predict edge importance, the reparameterization trick [21, 12] is adopted. Graphmask follows a similar framework but finds edge masks for each GNN layer. Zorro [7] identifies crucial nodes and node features using a greedy algorithm, while SubgraphX [42] employs Monte Carlo Tree Search [29, 13] to identify subgraph-level explanation. These methods incorporate a regularization term to limit the explanation size, but in our work, we explore finding a sparse explanation without relying on prior-knowledge-based regularization terms.

**Gradient/Features-Based, Decomposition, and Surrogate GNNs Explanation Methods**: Gradient/features-based methods such as [2, 24] utilize gradient or feature values to identify important nodes, edges, or node features, which are inspired by image explanation [43]. However, these methods ignore the structural information used by GNNs training, which makes them not proficient in finding topological explanations. Decomposition methods [27, 28] distribute the prediction score from the last layer to the input space, ignorance of activation functions limits the application of them. Two typical representative methods of surrogate methods are GraphLime [11] and PGMExplainer [35]. GraphLime extends the LIME framework [25] by training a nonlinear model in the N-hop neighborhood, while PGMExplainer builds a probabilistic graphical model to indicate node dependencies. Both approaches incorporate a regularization term to limit the size of the explanation like the aforementioned perturbation-based methods, which is the main obstacle we need to overcome in our work.

**Other works**: The work [32] focuses on both the explanation and complementary subgraphs. It requires the explanation subgraph to be sufficient for producing the same prediction as using the entire graph, and considers the complementary subgraph as a necessary condition, i.e., removing it leads to different predictions. We propose that the complementary subgraph contains no information about the original prediction, implying a uniform distribution as output if used as input. The work [30] and our work employ a similar idea of the uniform distribution, with their primary focus being the discovery of causal effects among features. This idea has inspired us to utilize an adversarial framework in conjunction with a scaling function to obtain a sparser explanation subgraph. There is also a line of research developed from the information bottleneck principle [41, 40, 23]. The intuition is that the explanation subgraph should contain enough information about the prediction, but also

**Table 1: Notations**

| Notation | Meaning |
|---|---|
| $G(V, E)$ | graph $G$ with vertex set $V$ and edge set $E$ |
| $G' \subseteq G$ | explanation set |
| $\overline{G'} \subseteq G$ | the complement subgraph of $G'$ w.r.t. $G$ |
| $I$ | mutual information |
| $\hat{Y}$ | predicted label |
| $U$ | uniform distribution |
| $mask \in [0, 1]^{n \times n}$ | informative weights of $n$ edges |
| $\odot$ | element-wise multiplication |

contain as less information about input graph as possible. In the works [41, 40, 23], hyperparameters are required to determine the final explanation subgraph, or to control the sparsity of the explanation set during the training phase, which is unknown in most real-world cases.

## 3 ADVERSARIAL MASK EXPLAINER

This work studies how to find an explanation for GNNs model's prediction, including node classification and graph classification. For the sake of simplicity in our discussion, we primarily focus on node classification in the subsequent sections. However, it is important to note that our approach is also applicable to graph classification and can be straightforwardly applied in practice. In this section, we begin by providing an overview of fundamental concepts related to GNNs. Subsequently, we formalize the problem of GNNs model explanation. Following that, we introduce our training objective function, specifically the adversarial objective. Moreover, we introduce a novel scaling function that effectively detects and enhances the mask weights of informative elements within the graph. Lastly, we present our innovative model, AMExplainer.

### 3.1 Preliminary

Given $G = (V, E)$ as the graph with the node set $V$ and the adjacent matrix $E$, the $mask$ is defined as a fraction matrix $mask \in [0, 1]^{n \times n}$, where $n = |V|$. When multiplied to $E$ element-wisely, $mask$ can be regarded as a small perturbation on the original graph $G$. We denote the perturbated new graph as $G' = G(V, E \odot mask)$, where $\odot$ is the element-wise multiplication. Then the problem we studied in this work is formalized as follows:

**DEFINITION 1 (PROBLEM OF GNNs EXPLANATION).** *Given a graph $G(V, E)$ and a trained GNNs model for node/graph classification, the objective is to identify a compact subgraph that can serve as the model's input while producing identical predictions to those obtained when the entire graph $G$ is used as input.*

The notations used in this work are summarized in Table 1.

### 3.2 Adversarial Objectives

To understand why the GNNs model makes a certain classification prediction, we target to find the most informative part of the input

graph, i.e., this *informative subgraph* should contain all the information to make a correct prediction. Existing works [39, 20] use the following objective function to look for a perturbed subgraph $G'$, which maximizes the mutual information $I$ between the predicted label $\hat{Y}$ and the perturbed graph $G'$:

$$G' = \arg\max_{G'} I(\hat{Y}, G') \qquad (1)$$

In addition, they incorporate a regularization term to constrain the size of $G'$ and curate the explanation set by selecting edges with the highest $K$ mask weights. The value of $K$ is manually determined based on prior knowledge. Interestingly, we get the following observation:

OBSERVATION 1. *If regarding the ground truth as the explanation subgraph, the remaining part of the entire graph (i.e., complementary subgraph) has no prediction ability on any class. The predicted classification distribution by the complementary subgraph is almost uniform.*

This is consistent with our intrinsic since we expect that the ground truth contains all the necessary knowledge that is required for accurate prediction, and the complementary subgraph contains only useless knowledge for prediction. **This inspires us to maximize the mutual information between the complementary subgraph and the uniform distribution to approach the elusive ground truth:**

$$G' = \arg\max_{G'} I(U, \overline{G'}) \qquad (2)$$

where $U$ is a uniform distribution, $G' = G(V, E \odot mask)$, and $\overline{G'} = G(V, E \odot (1 - mask))$. However, in practice, the GNNs model might not always succeed to find the ground truth to make the prediction. One of the reasons is that the accuracy of the trained GNNs model is not 100%. Another reason is that GNNs model exhibits a distinct cognitive logic and methodology in contrast to human cognition when it comes to comprehending graphs. This means that Eq 1 and Eq 2 can have different optimal values. To understand how the GNNs model make a certain prediction, we need Eq 1. To make the GNNs' cognitive logic on comprehending graphs understandable, we need Eq 2 to help the solution to approach the understandable ground truth. Hence, it naturally leads us to consider employing adversarial network techniques as a means to reconcile these two solutions. Following the training methodology of adversarial networks, the identical set of mask parameters is employed for cross-training the two objective functions. This adversarial process ensures the convergence of parameters towards an equilibrium point between the solutions of the two equations. The idea can be formalized as:

IDEA 1. *(1) Train the network with parameter mask using Eq 1 for $e_1$ epochs; (2) Train the network with parameter $1 - mask$ using Eq 2 for $e_2$ epochs; (3) Repeat (1&2) until the mask converge.*

We observe that the convexity of mutual information actually causes the *mask* used in cross-training to transition between solutions of Eq 1 and Eq 2. When Eq 1 is being trained, the *mask* moves towards the solution of Eq 1. When Eq 2 is being trained, the *mask* moves towards the solution of Eq 2. The final mask tends to converge to a position between the two solutions, and this position

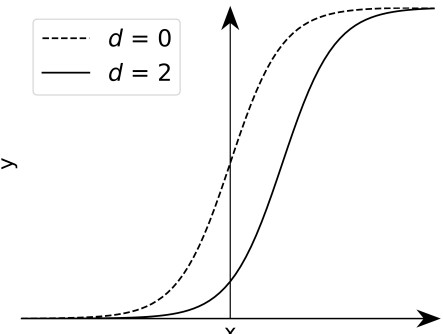

Figure 2: As $d$ **increases, the curve of** $Sigmoid(M, d)$ **moves from left to right.**

is actually adjusted by the epoch ratio $e_2/e_1$. While we leverage the concept of adversarial training, the fact that cross-training operates on the same set of parameters and both objective functions aim towards maximizing leads us to simplify the training process by combining the two objective functions into a single network. The new objective function is formulated as follows:

$$G' = \arg\max_{G'} [I(\hat{Y}, G') + \beta \cdot I(U, \overline{G'})] \qquad (3)$$

where $\beta$ is utilized to adjust the extremum point of this new convex function, which is formed as a linear combination of the two convex functions, namely Eq 1 and Eq 2. In other words, $\beta$ adjusts the equilibrium point of the solutions to Eq 1 and Eq 2, and its role is entirely equivalent to that of the epoch ratio $e_2/e_1$. Thus, we get our simplified idea:

IDEA 2 (EQUIVALENT BUT SIMPLIFIED). *Train the network with parameter mask using Eq 3.*

## 3.3 Scaling Function

After conducting extensive experiments, we found that training the network directly to optimize Eq 3 did not yield satisfactory results. However, we got the following two observations.

OBSERVATION 2. *During the training process, the mask weights of edges that contain substantial information and significantly contribute to accurate predictions* **rapidly increase** *and converge to values close to 1.*

OBSERVATION 3. *During the training process, the mask weights of edges that carry irrelevant information and provide little assistance in making accurate predictions* **decrease slowly** *and tend to remain far from approaching zero.*

To enhance training efficiency, these two observations inspired us to prioritize maintaining mask weights close to 1 for important edges while actively suppressing the mask weights of irrelevant edges. This strategy aims to expedite the convergence of mask weights associated with irrelevant edges towards zero, ensuring the sparsity of the explanation set. In general, the sigmoid function is

---

**Algorithm 1** AMExplainer

**Input:** The input graph $G = (V, E)$, a trained GNNs model, number of epochs $n$, interval epochs $T$, shift step $D$, $\beta$, and initialized $mask$.

**Output:** The explanation set.

1: **for** $i = 1, ..., n$ **do**
2:    **if** $i$ is divisible by $T$ **then**
3:       Increase the value of $d$ in the $Sigmoid(mask, d)$ by $D$;
4:    $mask = Sigmoid(mask, d)$;          ▷ Eq 4
5:    Compute loss;                  ▷ Eq 3
6:    Update $mask$ by backpropagation;
7: Include all the edges whose $mask \geq 0.1$ into the explanation set.

---

commonly employed for re-scaling mask values. However, in light of our Observation 2 and Observation 3, we require the sigmoid function to possess a capability of rightward shifting. This ensures that the weights of important edges remain close to 1, while the weights of irrelevant edges fall within the steepest region of the sigmoid function and converge to zero more rapidly. In this paper, we introduce a novel scaling function defined as a sigmoid function with a shift parameter:

$$Sigmoid(M, d) = \frac{1}{1 + \exp\left(-(M - d)\right)} \quad (4)$$

where $M$ is the element of the mask matrix, and $d$ is the shift parameter. As illustrated in Figure 2, the curve shifts towards the right as $d$ increases, effectively driving the weights of irrelevant edges towards zero. Lastly, and perhaps most importantly, it is crucial for the shift parameter to gradually increase from 0. This ensures that the weights of important edges do not mistakenly fall within the steepest region of the sigmoid function before reaching proximity to 1, which would result in their erroneous exclusion from the explanation set.

### 3.4 Adversarial Mask Explainer: AMExplainer

The mask values were expected to follow a random distribution within the range of [0, 1]. However, in the actual experiments, we observed that the majority of edges had mask values that were nearly close to zero, while only a small fraction of edges fell within the range of [0.1, 1]. After completing the mask training process, in order to avoid missing any potentially informative edge, we adopt a very conservative edge selection strategy. Specifically, we set the threshold for including edges in the explanation set to a very low value of 0.1. If the mask value obtained from training for a particular edge exceeds 0.1, it is included in the explanation set.

Finally, AMExplainer is formalized as Algorithm 1. The $mask$ is updated every epoch by backpropagation, and the shift parameter increases from 0 with a step $D$ every $T$ epochs.

### 3.5 Regularization Term *vs.* Eq 2

Some may argue that Eq 2 serves as a regularization term in Eq 3, aiming to suppress the magnitude of mask values. However, the reality is that the size of the explanation set is unknown, and thus, it is inappropriate for algorithm designers to impose a predetermined

constraint on the size of the explanation set based on personal preferences. The true role of Eq 2 is to facilitate a more interpretable explanation that closely resembles a ground truth interpretation by incorporating it into the adversarial process with Eq 1. The sparsity of the results, in fact, emerges as an incidental outcome of pursuing the interpretability of solutions.

## 4 EXPERIMENTAL STUDIES

In this section, we conduct extensive experiments to study two essential research questions:

1. To what extent AMExplainer can provide an effective explanation set?
2. How sparse is the explanation set?

We study the two research questions for both node classification and graph classification. The baselines are chosen to be GNNExplainer [39] and PGExplainer [20]. In the following, we first introduce the benchmark datasets used in this work. Then we give the setup details for reproducibility. Finally, we answer the aforementioned two research questions by the experimental results.

### 4.1 Baselines

We compare AMExplainer with the following baselines:

- **GNNExplainer[39]:** The first model-agnostic approach to providing explanations for the output of a trained GNNs model. It learns masks over node features and edges. For comparison with AMExplainer, we use GNNExplainer to learn masks over edges.
- **PGExplainer[20]:** This approach trains a parameterized mask predictor to generate masks over edges. The predictor takes edge embeddings as input, which are the concatenation of node embeddings, and outputs the probability of each edge being selected.
- **GSAT[23]:** GSAT learns masks over edges based on the information bottleneck principle. The selected edges should be informative for the original prediction while containing limited information from the input.
- **OrphicX[18]:** From a causal perspective, this approach identifies informative edges by determining the causal components in the latent space that contribute to the prediction.

### 4.2 Dataset[39, 20]

For the node classification task, we use four synthetic datasets to test the performance of our algorithm:

1. **BA-Shapes:** This dataset comprises a single base Barabási-Albert (BA) graph consisting of 300 nodes. Additionally, 80 house-structured motifs are randomly attached to the BA graph, followed by the introduction of additional edges to perturb the overall graph. Nodes in the base BA-graph are assigned the same label, while nodes in the house-structured motif are classified into three distinct categories based on their relative positions within the *house* structure (top, middle, or bottom).
2. **BA-Community:** The BA-Community dataset bears resemblance to the BA-Shapes dataset, as it connects two BA-Shapes while incorporating node features sampled from two

distinct Gaussian distributions. Nodes with features sampled from the same Gaussian distribution are deemed to belong to the same community, along with their varying relative positions within each BA-Shape. This combination results in a total of 8 classes in the BA-Community dataset.

3. **Tree-Cycles:** The Tree-Cycle dataset utilizes a balanced binary tree as the underlying graph, supplemented by the random attachment of 80 cycle-structured motifs to the tree structure. This dataset exhibits two distinct types of labels, depending on whether the node is located within the base tree graph or not.

4. **Tree-Grid:** The Tree-Grid dataset shares similarities with the Tree-Cycle dataset, with the distinction being the utilization of grid-structured motifs instead of cycle-structured motifs. The label assignment follows the same setup as the Tree-Circles dataset.

For the graph classification task, we use one synthetic dataset and one real-world dataset to test the performance of our algorithm:

5. **BA-2motifs:** The synthetic dataset, BA-2motifs, comprises 800 graphs, half of which are augmented with house-structured motifs, while the remaining half are appended with five-node-cycle motifs. Consequently, the BA-2motifs dataset encompasses two distinct classes based on the type of attached motifs.

6. **Mutag:** The Mutag dataset is a real-world dataset consisting of 4337 graphs categorized into two classes. A graph is labeled as mutagenic if it contains the chemistry groups $NH_2$ or $NO_2$, while graphs without these groups are considered nonmutagenic.

## 4.3 Setup and Reproducibility

**Mean Squared Error:** These studies [39, 20] provide evidence that, within the scope of our investigation, mutual information can be effectively replaced by cross-entropy. Cross-entropy is usually employed as a metric to quantify the divergence between two probability distributions. In the context of our research problem, it serves to gauge the disparity between the class probability distribution predicted using the original graph and the probability distribution predicted using only the informative subgraph. In the practical training procedure, we discovered that substituting cross-entropy with Mean Squared Error (MSE) yields comparable or even improved training outcomes. Consequently, in real-world applications, we propose the direct utilization of MSE as a replacement for mutual information in the two loss terms during training.

**Assessment of Effectiveness:** To evaluate the effectiveness of the AMExplainer in comparison to the baselines, we assess the extent to which the prediction distribution generated by the explanation set resembles that produced by the entire graph. For this evaluation, we employ the measure of absolute fidelity, denoted as $fidelity_{abs}$:

$$fidelity_{abs} = \frac{1}{N} \sum_{i=1}^{N} |p_{y_i} - p_{m_i}| \quad (5)$$

Here $i$ is the node index. $p_{y_i}$ is the original prediction probability of assigning node $i$ into the correct class by the entire graph, and $p_{m_i}$ is the prediction probability of assigning node $i$ into the correct class

by the subgraph induced by the explanation set. The expression $\frac{1}{N} \sum_{i=1}^{N}$ represents the average taken over all nodes within a motif, where $N$ denotes the number of nodes contained in the motif. A lower value of $fidelity_{abs}$ indicates a closer similarity between the prediction distribution generated by the explanation set and that produced by the original graph. This, in turn, signifies higher effectiveness of the algorithm.

One might question why we chose fidelity as the measure of effectiveness. Since the task at hand is classification, why not use a metric such as the following: if the informative subgraph and the entire graph predict the same label when used as input to GNNs, the metric is 0; otherwise, the metric is 1. In reality, this seemingly stricter metric compared to fidelity harbors an inherent pitfall in its underlying logic. Specifically, predicting the same label does not necessarily imply that GNNs have utilized the same informative graph at the lower levels. For instance, consider a scenario where the informative subgraph and the entire graph, when used as input to GNNs, yield prediction probabilities of 51% and 99%, respectively, for a certain label. Although both cases yield the same label prediction, it is evident that in this situation, the informative graph/explanation set used in the case of a fluctuating prediction at 51% and an extremely confident prediction at 99% would certainly differ significantly. Considering this scenario, we have chosen fidelity as a more reasonable measure specifically tailored for the classification task in this work.

**Assessment of Sparsity:** One crucial distinction between human reasoning and machine reasoning when seeking explanations for a problem lies in the difficulty of humans constructing complex sets of causes, akin to machines. Humans tend to simplify problems by reducing them to a few fundamental causes. In other words, we desire the algorithm to yield a sparse explanation set, even though the algorithm designers themselves may not be certain whether the true explanation set is indeed sparse. It is not appropriate to impose the designer's subjective preference for sparsity on the algorithm. However, extensive experiments have demonstrated that the sparsity of the solution can be naturally achieved during the process of employing Eq 2 to progressively align the mask with the ground truth. In this work, we define the $sparsity$ as:

$$sparsity = \frac{1}{N} \sum_{i=1}^{N} (1 - \frac{|m_i|}{|M_i|}) \quad (6)$$

Here $i$ is the node index. The term $|m_i|$ refers to the size of the explanation set for node $i$, while $|M_i|$ represents the total number of edges in the computation graph. The expression $\frac{1}{N} \sum_{i=1}^{N}$ represents the average taken over all nodes within a motif, where $N$ denotes the number of nodes contained in the motif. A higher degree of sparsity implies a sparser explanation set, which in turn indicates that the explanation set is more easily comprehensible by human reasoning.

## 4.4 Fidelity

**Node Classification Task:** Table 2 presents the average fidelity of the results obtained when five different algorithms are applied to perform node classification tasks on four distinct datasets. For instance, we executed the AMExplainer algorithm on all five nodes within a specific motif from the BA-shapes dataset. Subsequently,

Table 2: Fidelity of node classification

| Algorithm | BA-shapes | BA-Com | Tree-Cycle | Tree-Grid |
|---|---|---|---|---|
| AMExplainer | 0.0028 | 0.0342 | 0.0072 | 0.0015 |
| GNNExplainer | 0.1743 | 0.1704 | 0.1535 | 0.0389 |
| PGExplainer | 0.5439 | 0.6005 | 0.0402 | 0.0253 |
| GSAT | 0.0010 | 0.0017 | 0.0079 | 0.0034 |
| OrphicX | 0 | 0 | 0 | 0 |

Table 3: Fidelity of graph classification

| Algorithm | BA-2motifs | Mutag |
|---|---|---|
| AMExplainer | 0.0144 | 0.0001 |
| GNNExplainer | 0.2625 | 0.1748 |
| PGExplainer | 0.2492 | 0.2669 |
| GSAT | 0.0073 | 0.0010 |
| OrphicX | 0 | 0 |

we calculated the fidelity based on Eq 5 and repeated the process by randomly selecting motifs from the entire graph, computing their average fidelity using the same approach. Then we get the first data 0.0028 in Table 2. AMExplainer further reduces the fidelity compared to GNNExplainer and PGExplainer by 79.93% to 98.39%. Although GSAT and OrphicX include all edges in their final explanations, as discussed in the next section, AMExplainer's fidelity performance remains comparable. This suggests that the explanation set produced by AMExplainer closely aligns with the original prediction. This implies that the explanation set generated by AMExplainer is close to the true subgraph used by the underlying logic of GNNs for node classification prediction.

**Graph Classification Task:** the computation of fidelity differs from node classification tasks. Since graph classification predictions are made on a graph level rather than on a per-node basis, there is no need to take an average over different nodes. Instead, fidelity is calculated by randomly selecting different graphs from the dataset and averaging the results. Compared to the better-performing approach between GNNExplainer and PGExplainer, AMExplainer further reduces fidelity by 94.22% to 99.94%. A comparable performance is achieved by AMExplainer when compared with GSAT and OrphicX, which take input as the explanation. When applied to the real-world dataset Mutag, AMExplainer demonstrates its best performance. When GNNs take the subgraph generated by AMExplainer as input, the resulting graph classification probabilities are nearly identical to those obtained when the entire original graph is used as input, with a difference of around 0.01%.

## 4.5 Sparsity

**Node Classification Task:** Now we compare the sparsity of the explanation obtained by AMExplainer with baselines. In fact, the concept of sparsity does not directly apply to these baselines because they select the top-$K$ edges based on the mask values as the explanation set, which means the size of the explanation set is fixed. To ensure fairness, we still count the number of edges with mask values greater than 0.1 and calculate the sparsity using the same method as in Equation 6. The experimental results are presented in Table 4.

We can see that three baselines, PGExplainer, GSAT, and OrphicX, exhibit a sparsity of 0 across all datasets. This is because they include all edges in the explanation set when the mask value threshold is set to 0.1. In comparison, the sparsity of GNNExplainer provides more meaningful insights. In relation to GNNExplainer, AMExplainer demonstrates a decrease in sparsity by approximately 20% on the BA-shapes and Tree-Grid datasets, an increase in sparsity by approximately 20% on the Tree-Cycle dataset, and comparable

performance to GNNExplainer on the BA-Community dataset. Experimental results indicate that while AMExplainer significantly improves prediction effectiveness by 1 to 2 orders of magnitude, it also maintains a reasonable level of sparsity in the explanation set, comparable to the baselines.

**Graph Classification Task:** In the case of graph classification tasks, as shown in Table 5, we observe that three baselines, PGExplainer, GSAT, and OrphicX, assign values greater than 0.1 to all masks, rendering the discussion of sparsity for them meaningless. When compared to GNNExplainer, AMExplainer exhibits a slight decrease in sparsity on the Mutag dataset, approximately 20%, while showing an increase in sparsity of about 60% on the BA-2motifs dataset. Overall, AMExplainer performs on par with or slightly better than the baselines in terms of solution sparsity.

## 4.6 Ablation study

**Scaling Function:** An ablation study was conducted to independently assess the practical impact of the scaling function. In other words, we investigated whether it is necessary to shift the scaled curve to the right during the training process. Results are shown in Table 6. In this experiment, we set the value of $d$ to 0 and used node classification as an example to observe the comparison between the experimental results and those obtained with the normal setting of $d$. The numerical values following the slash represent the experimental results obtained under the regular setting of $d$. We observed that the setting of $d$ had a minor impact on fidelity but a significant effect on sparsity. The scaling function shifted to the right enhanced sparsity by 1.7 to 7 times compared to a regular sigmoid function. In other words, the scaling function effectively removed ineffective information from the explanation set, achieving a sparse explanation set while ensuring prediction accuracy.

**Extremum Point:** As we can adjust the extremum point of the convex function (Eq 3) using $\beta$, we conducted another ablation study to examine the effect of the second term in Eq 3. We set the value of $\beta$ to 0, so the loss function only included the first term. In other words, during training, we did not consider forcing the prediction distribution of the complement set of the explanation set to approximate a uniform distribution. Using node classification as an example, the results are shown in Table 7. The numerical values following the slash represent the experimental results obtained under the regular setting of $\beta$. We observed a slight decrease in sparsity, ranging from 5% to 20%, after adding the second term to the loss function. However, fidelity experienced a significant improvement, ranging from 2 to 119 times. In other words, AMExplainer achieves a significant enhancement in prediction accuracy by sacrificing a small degree of sparsity in the explanation set. Through these two

**Table 4: Sparsity of node classification**

| Algorithm | BA-shapes | BA-Community | Tree-Cycle | Tree-Grid |
|---|---|---|---|---|
| AMExplainer | 0.6475(↓ 26.96%) | 0.8745(↓ 3.13%) | 0.8397 (↑ 19.14%) | 0.7509(↓ 17.43%) |
| GNNExplainer | 0.8865 | 0.9028 | 0.7048 | 0.9094 |
| PGExplainer | 0 | 0 | 0 | 0 |
| GSAT | 0 | 0 | 0 | 0 |
| OrphicX | 0 | 0 | 0 | 0 |

**Table 5: Sparsity of graph classification**

| Algorithm | BA-2motifs | Mutag |
|---|---|---|
| AMExplainer | 0.4509(↑ 58.27%) | 0.7335(↓ 19.63%) |
| GNNExplainer | 0.2849 | 0.9126 |
| PGExplainer | 0 | 0 |
| GSAT | 0 | 0 |
| OrphicX | 0 | 0 |

**Table 6: Ablation study of AMExplainer with different $d$**

| | BA-shapes | BA-Com | Tree-Cycle | Tree-Grid |
|---|---|---|---|---|
| Fidelity | 0.0046/0.0028 | 0.0284/0.0342 | 0.0021/0.0072 | 0.0158/0.0015 |
| Sparsity | 0.3895/0.6475 | 0.3661/0.8745 | 0.1421/0.8397 | 0.2076/0.7509 |

**Table 7: Ablation study of AMExplainer with different $\beta$**

| | BA-shapes | BA-Com | Tree-Cycle | Tree-Grid |
|---|---|---|---|---|
| Fidelity | 0.3333/0.0028 | 0.1977/0.0342 | 0.0171/0.0072 | 0.0101/0.0015 |
| Sparsity | 0.8376/0.6475 | 0.9163/0.8745 | 0.9861/0.8397 | 0.9643/0.7509 |

sets of ablation studies, we found that the scaling function effectively removes redundant information to achieve sparsity while ensuring prediction accuracy. Additionally, the second term in Eq 3 significantly enhances prediction accuracy while maintaining sparsity essentially unchanged.

### 4.7 Summary

Now we can answer the two research questions that were posed at the beginning of this section:

1. When applied to node classification tasks, AMExplainer outperforms the state-of-the-art by reducing fidelity by 79.93%. In other words, AMExplainer achieves an effectiveness improvement of 1 order of magnitude compared to the state-of-the-art [39]. Here we only consider GNNExplainer because it is meaningless to compare AMExplainer with baselines that include all the edges into the explanation set with information redundancy. For graph classification tasks, AMExplainer surpasses the state-of-the-art by reducing fidelity by 94.22%. This implies an effectiveness improvement of 2 orders of magnitude compared to the state-of-the-art.

2. In both node classification and graph classification tasks, the sparsity of the explanation set generated by AMExplainer is comparable to that of the state-of-the-art, with fluctuations of around 20% in both directions. Overall, while significantly enhancing prediction effectiveness by 1 to 2 orders of magnitude, AMExplainer maintains a similar level of sparsity to the state-of-the-art, ensuring its effectiveness matches or exceeds that of the existing approaches.

## 5 CONCLUSION AND FUTURE WORK

In this paper, we have studied the problem of explainability in GNN models. Leveraging the concept of adversarial networks, we have proposed a novel explainability model called AMExplainer. The objective of this model is twofold: to find an informative graph that provides accurate prediction probabilities and to find an informative graph that aligns with the understanding of humans, resembling a ground truth. During the training process, these two objectives create an adversarial relationship, resulting in a balanced solution that captures both aspects. Additionally, we have introduced a novel scaling function to expedite the convergence of the solution during training. Extensive experiments have demonstrated that AMExplainer outperforms the state-of-the-art by improving prediction effectiveness by 1 to 2 orders of magnitude while preserving the sparsity of the solution. Moreover, this performance improvement is particularly prominent on real-world dataset.

The AMExplainer, while it is promising for generating explanation subgraphs, faces a limitation. The issue lies in its inability to achieve a fully sparse explanation subgraph solely by controlling the move distance of the sigmoid function. Ideally, we aim for a scenario where all informative edges are assigned a mask value of 1, while all irrelevant edges receive a mask value of 0. Our experimental results indicate that there exists a small subset of edges with mask values falling between 0.1 and 1, suggesting that the current approach falls short of achieving complete sparsity. To address this challenge, a potential solution lies in introducing adjustments to the slope of the sigmoid function during the training phase. By making the sigmoid function steeper, we can exert greater control over the mask values, effectively driving them toward the desired extremes of 0 and 1. This approach requires a carefully designed schedule for modifying both the move distance and slope rate of the sigmoid function. We leave this part for future work. It is important to highlight that while AMExplainer is primarily tailored for elucidating the inner workings of a trained GNNs model, its utility extends beyond GNNs and can be effectively employed with various other types of neural networks.

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
