# OpenReview forum: "Adversarial Mask Explainer for Graph Neural Networks"
_ACM.org/TheWebConf/2024/Conference — TheWebConf24 Oral_

### Official Review · Reviewer_6y4Z · 2023-11-21

**Novelty:** 4
**Technical Quality:** 5

**Review:**

This paper studies the problem of instance-level explanations for GNNs. The authors claim the previous methods rely on the predefined explanation size, and propose the AMExplainer, which leverages the scaling function to automatically select the desired subgraphs.

Pros:
1. The problem of finding instance-level explanations for GNNs is interesting and worth exploring.
2. The proposed method can automatically choose the size of the explanation subgraph while maintaining sparsity.
3. Experimental results demonstrate that the proposed method can reduce the fidelity in both node classification tasks.

Cons:
1. The motivation that complementary subgraph has no prediction ability on any class is unclear.
2. The graphs used in the experiments are quite small and specific. Can the proposed method provide explanations for the general node classification task?
3. The authors leverage fidelity as the metric for both node classification and graph classification. However, methods such as GSAT and OrphiX can achieve near 0 fidelity, alough their spasity might be 0. Does the fidelity a good metric for this task?

**Questions:**

1. Where does observation 1 come from?
2. For the node classification tasks, how do you select the number of layers of GNNs? For example, does the node beyond the k-hop of the target node belong to the explanation subgraph if the GNN only has k layers?
3. How do you calculate the mutual information between the complementary subgraph and the uniform distribution for the node classification task? If the target node's first-hop neighbors all belong to the explanation subgraph, the complementary graph would isolate the target node.

**Reviewer Confidence:**

3: The reviewer is confident but not certain that the evaluation is correct

**Scope:**

3: The work is somewhat relevant to the Web and to the track, and is of narrow interest to a sub-community

---

### Official Review · Reviewer_tpH9 · 2023-11-23

**Novelty:** 3
**Technical Quality:** 3

**Review:**

The paper studies the problem of explainability of graph neural networks (GNN). The paper proposes a novel approach for explaining graph neural networks by identifying an explanation set whose complement set consists of uninformative edges, and then employing a scaling function go further remove additional uninformative edges.

The paper provides a nice motivation and clear introduction to the topic of explainability for GNNs, and the challenges of using standard approaches. However, the paper is not clear in several parts (see below), and the originality is somehow limited, since at the end the approach is very similar to a regularized version of a problem of selecting a small subgraph with performance similar to the full model, that has been proposed before. Moreover, the experimental evaluation considers only 1 real dataset and includes synthetic datasets where very simple explanation are planted in the graph, making it difficult to assess the significance of the approach in practice.

PROS
- The abstract and introduction provide a nice motivation and clear introduction to the topic
- Overall, the idea of using as a second objective the mutual information of the complementary subgraph and the uniform distribution is interesting
- The experimental results show that the proposed method works better than previous approaches on the datasets considered

CONS
- Several parts of the paper are not clear, here is a (partial) list.

Observation 1is not clearly written; please rewrite.

Line 308: “with our intrinsic since” what is an “intrinsic”?

Lines 381-382 (“Another reason is that … to comprehending graphs”) is entirely unclear; what is the “cognitive logic and methodology” of a GNN?

Idea 2: there is no proof that the solution of equation 3 is equivalent to the solution in idea 1.

How is the beta parameter set?

In section 3.4: the selection of threshold 0.1 for including an edge in the explanatory set is in contrast with observation 2 (such edges show converge to values close to 1).
- While the method is described in terms of mutual information, at the end the approach uses mean square error. At this point, it is unclear why the description is not made in terms of MSE, or why the two notions should be equivalent.
- From equation 3, at the end the approach is very similar to a regularization approach, where beta governs the sparsity of the solution (that does not derive from pursuing the interpretability of solutions, but it is enforced by the algorithm).
- The code is not available, and there is no mention of whether it will be made available if the paper is accepted

**Questions:**

- Can you address the comments in the first point of CONS (with at most 3 sentence each)?
- What is the value of describing the method in terms of mutual information, if at the end MSE is used?
- How is the beta parameter set in your experiments? Is this different from fixing a regularization parameter with (cross-)validation?
- Can you provide an anonymous repository with the code for the method and to reproduce the results?

**Reviewer Confidence:**

3: The reviewer is confident but not certain that the evaluation is correct

**Scope:**

4: The work is relevant to the Web and to the track, and is of broad interest to the community

---

### Official Review · Reviewer_q6Re · 2023-11-24

**Novelty:** 4
**Technical Quality:** 5

**Review:**

This paper focuses on the issue of graph neural network interpretability. Specifically, the authors utilize the concept of adversarial networks to implement a dual optimization objective in the objective function to ensure accurate prediction of the mask and sparsity of the interpretation set. And a scaling function is designed to automatically sense and scale the weights of the information part of the graph, which filters out unimportant edge/node/node features to speed up the convergence of the solution during the training process.

Strengths:
1.	This paper focuses on an important issue of explaining graph neural networks.

2.	The proposed method is reasonable and technically sound.

3.	The results are promising when compared to representative perturbation-based interpreters.

Weaknesses:

1.	Some of the author's statements are confusing, for example, in line 160, "the complement set of the explanation edge set (indicated by solid lines, while the complement set is marked by dotted lines)", whether the solid lines indicate the explanation edge set or the complement set?

2.	As far as I know, there are some decomposition methods to explain graphical neural networks that have also achieved good results. However, in the experimental section, the authors do not compare the decomposition methods with the proposed framework. It is suggested to add a decomposition-based interpreter as a baseline.

3.	Whether the authors attempted node classification experiments on a wider range of real datasets such as Cora, PubMed, ogbn-arxiv, and others.

4.	It is suggested to add some visualizations to show more clearly the interpretability of the proposed method.

5.	Given the relative complexity of the proposed methodology, a temporal and spatial complexity analysis or an empirical speed assessment would be useful in assessing its significance.

**Questions:**

Please refer to the weaknesses.

**Reviewer Confidence:**

4: The reviewer is certain that the evaluation is correct and very familiar with the relevant literature

**Scope:**

3: The work is somewhat relevant to the Web and to the track, and is of narrow interest to a sub-community

---

### Official Review · Reviewer_8PCX · 2023-11-27

**Novelty:** 3
**Technical Quality:** 3

**Review:**

This paper proposed a simple dual optimization target function by leveraging the concept of adversarial networks to generate a sparse set of masks with high fidelity in order to explain the predictions made by GNNs.

**Questions:**

1. When explaining predictions made by GNNs, Explanation Accuracy is commonly used as an evaluation metric. However, this paper deviates from this standard and employs an alternative metric, Absolute Fidelity. What are the advantages of using this metric? Additionally, the paper should incorporate Inference Time to assess model efficiency.

2. It is advisable to consider GRAD and ATT as additional baselines.

3. Could the authors please provide specifics on how the mutual information between label and graph, as described in Eq 1, is calculated?

4. The temperature parameter $\beta$ is crucial, and the authors should investigate how varying its values across a broader range affects the experimental results.

5. What is the hyperparameter K mentioned in the abstract?

**Reviewer Confidence:**

2: The reviewer is willing to defend the evaluation, but it is likely that the reviewer did not understand parts of the paper

**Scope:**

4: The work is relevant to the Web and to the track, and is of broad interest to the community

---

### Official Review · Reviewer_bHiH · 2023-11-30

**Novelty:** 3
**Technical Quality:** 2

**Review:**

This paper introduces Adversarial Mask Explainer (AMExplainer), a framework for achieving instance-level explainability in Graph Neural Networks (GNNs). The authors elucidate the significance of each node in the graph by generating a set of masks, noting that the complement of these masks serves as an indicator of a selected node's importance. AMExplainer employs adversarial networks to optimize both accurate mask prediction and the sparsity of the explanation set. Additionally, it introduces a scaling function to amplify the weights of the informative part of the graph, enhancing convergence. The method has been experimentally proven to be accurate and interpretable, applicable to various downstream tasks. Experimental results demonstrate that AMExplainer generates a sparse set of masks, significantly improving prediction effectiveness compared to previous methods. In general, this paper  presents a novel approach to explainability in GNNs that addresses a key challenge in the field. The proposed method is well-motivated and well-evaluated. From the description of the paper, it seems that the authors utilize a set of masks to conduct node classification prediction with the help of adversarial network and scaling function. The topic discussed is still far away from explainability of the GNNs.

**Questions:**

- Can the authors provide more theoretical results about the explainability not expressing them just by word.

- Could the authors provide details about the selection of the hyperparameters? Is the model robust to different hyperparameter settings?

- Could the authors briefly describe the mask initialization process?

**Reviewer Confidence:**

2: The reviewer is willing to defend the evaluation, but it is likely that the reviewer did not understand parts of the paper

**Scope:**

3: The work is somewhat relevant to the Web and to the track, and is of narrow interest to a sub-community

---

### Decision · Program_Chairs · 2024-01-22

**Decision:**

Accept (Oral)

**Comment:**

The reviewers appreciate the submission's contribution in terms of novel methods and new angles to tackle the tradeoff between sparsity and explanation faithfulness. The approach is also sound and justified.
 The reviewers have raised a number of concerns, including consideration of baselines; hyperparameters; confusing writing; code. Most of the concerns are addressed.